# Pyruvate Dehydrogenase Kinase Inhibition by Dichloroacetate in Melanoma Cells Unveils Metabolic Vulnerabilities

**DOI:** 10.3390/ijms23073745

**Published:** 2022-03-29

**Authors:** Jiske F. Tiersma, Bernard Evers, Barbara M. Bakker, Mathilde Jalving, Steven de Jong

**Affiliations:** 1Department of Medical Oncology, University of Groningen, University Medical Center Groningen, Hanzeplein 1, 9713 GZ Groningen, The Netherlands; j.f.tiersma@umcg.nl; 2Laboratory of Pediatrics, Section Systems Medicine of Metabolism and Signalling, University of Groningen, University Medical Center Groningen, Antonius Deusinglaan 1, 9713 AV Groningen, The Netherlands; b.evers@umcg.nl (B.E.); b.m.bakker01@umcg.nl (B.M.B.)

**Keywords:** melanoma, metabolism, dichloroacetate, metabolic reprogramming

## Abstract

Melanoma is characterized by high glucose uptake, partially mediated through elevated pyruvate dehydrogenase kinase (PDK), making PDK a potential treatment target in melanoma. We aimed to reduce glucose uptake in melanoma cell lines through PDK inhibitors dichloroacetate (DCA) and AZD7545 and through PDK knockdown, to inhibit cell growth and potentially unveil metabolic co-vulnerabilities resulting from PDK inhibition. MeWo cells were most sensitive to DCA, while SK-MEL-2 was the least sensitive, with IC_50_ values ranging from 13.3 to 27.0 mM. DCA strongly reduced PDH phosphorylation and increased the oxygen consumption rate:extracellular acidification rate (OCR:ECAR) ratio up to 6-fold. Knockdown of single PDK isoforms had similar effects on PDH phosphorylation and OCR:ECAR ratio as DCA but did not influence sensitivity to DCA. Growth inhibition by DCA was synergistic with the glutaminase inhibitor CB-839 (2- to 5-fold sensitization) and with diclofenac, known to inhibit monocarboxylate transporters (MCTs) (3- to 8-fold sensitization). CB-839 did not affect the OCR:ECAR response to DCA, whereas diclofenac strongly inhibited ECAR and further increased the OCR:ECAR ratio. We conclude that in melanoma cell lines, DCA reduces proliferation through reprogramming of cellular metabolism and synergizes with other metabolically targeted drugs.

## 1. Introduction

There is still a lack of effective treatments in patients with metastatic melanoma who do not benefit sufficiently from treatment with immunotherapy [1,2]. In cutaneous melanoma, mitogen-activated protein kinase (MAPK) signaling is frequently enhanced due to activating mutations in BRAF (~50%), NRAS Q61 (15–28%) or NF1 (14%) [3,4]. Melanoma patients with an activating mutation in the BRAF oncogene can be treated with BRAF inhibitors, combined with inhibitors of the downstream MAPK kinase (MAPKK or MEK). However, the duration of response is limited mainly due to the reactivation of the MAPK pathway [5]. Other mechanisms of resistance to BRAF inhibitors include metabolic adaptations such as increased oxidative metabolism, decreased lactate secretion and increased dependency on glutamine [6,7]. For BRAF-wildtype melanomas and for BRAF-mutated melanoma, especially those that have become resistant to BRAF inhibitors, targeting the downstream consequences of MAPK pathway activation is of interest.

Independent of the specific mutation, the activation of the MAPK pathway results in increased stability of hypoxia inducible factor 1 (HIF1) in both normoxia and hypoxia [8]. Accumulation of HIF1 results in the upregulation of genes involved in glucose uptake, and melanoma cells of varying oncogenic backgrounds display highly glycolytic phenotypes. In melanoma cells in vitro, 60–90% of glucose is converted to lactate, contributing to metabolic reprogramming, supporting the generation of ATP and building blocks for cell division and maintaining redox balance [9,10]. Direct targeting of HIF has been disappointing, but the targeting of downstream targets of HIF such as the pyruvate dehydrogenase kinases (PDKs) is of interest [11,12]. There are four isoforms (PDK1-4), and all isoforms phosphorylate pyruvate dehydrogenase (PDH), thereby inactivating it. Active PDH catalyzes the conversion of pyruvate into acetyl CoA, which in turn enters the citric acid cycle to enable oxidative phosphorylation (OXPHOS) [13]. Phosphorylated, inactive PDH contributes to the glycolytic phenotype, making PDK a potential target in melanoma. Indeed, PDK1 and PDK2 are upregulated in human melanoma compared to benign nevi and are therefore interesting therapeutic targets [14]. 

Dichloroacetate (DCA) is a pan-PDK inhibitor and induces a switch in cellular metabolism towards OXPHOS, accompanied by a secondary reduction in glycolysis. DCA has been used clinically since the 1980s for the treatment of lactic acidosis, mitochondrial defects and PDH deficiency [15,16]. To date, four small (N = 5–23) early phase clinical trials have tested DCA in patients with several types of advanced solid tumors, and the recommended phase II dose has been determined [17,18,19,20]. One very small phase II study demonstrated tumor stabilization or regression in 4 out of 5 glioblastoma patients [19]. A more recently developed PDK inhibitor is AZD7545, which has a higher specificity for PDK1 and 2 than DCA and is therefore effective at lower concentrations. However, it has not been tested in patients [21]. 

We hypothesized that targeting glycolysis through PDK inhibition will reverse metabolic reprogramming in melanoma cell lines and reduce cell growth. Here, we determine the effect of PDK inhibition using the drugs DCA and AZD7545 or by PDK downregulation on viability, expression of metabolic enzymes, and cellular metabolic function in a panel of melanoma cell lines with various genetic backgrounds. However, tumor cell growth is usually not heavily affected by metabolic inhibition with a single agent due to metabolic plasticity, which results in the activation of compensatory pathways. This may require dual metabolic targeting [22,23]. Therefore, we investigate whether PDK inhibition results in increased vulnerabilities of melanoma cells for other metabolically targeted drugs. 

## 2. Results

### 2.1. PDK Inhibition Inhibits Proliferation in 2D and 3D Culture

All melanoma cell lines in our panel exhibited sensitivity to DCA (Figure 1A). The MeWo and A375 (BRAF^V600E^) cell lines showed comparable sensitivity to DCA with IC_50_ values of 13.3 and 14.9 mM, whereas the SK-MEL-2 (NRAS^Q61R^) and SK-MEL-28 (BRAF^V600E^) cell lines were much less sensitive to DCA (Table 1). Sodium acetate, used at the same concentrations as DCA, did not affect the viability. Established A375 spheroids, grown for three days prior to initiation of treatment for 96 h, had an IC_50_ value for DCA of 29.3 ± 6.1 mM (Figure 1B), whereas A375 spheroids that were treated directly after seeding for 7 days had an IC_50_ value of 14.2 ± 1.5 mM (Figure 1C). Seven-day treatment of spheroids with 35 mM DCA completely inhibited cell growth, and no spheroids were formed (Figure 1D). In contrast, short-term (last 24 h) treatment of established spheroids with DCA did not markedly interfere with spheroid integrity or viability, as reflected by positive calcein staining of all spheroids. 

### 2.2. DCA Decreases PDH Phosphorylation in Melanoma Cell Lines

Next, we investigated the effect of DCA on PDH phosphorylation. There are three phosphorylation sites in the PDH E1α subunit: site 1 (Ser293), site 2 (Ser300) and site 3 (Ser232) (Figure 2A). Of these sites, phosphorylation of pPDH^293^ is the most rapid and the most well-known mechanism of PDH inactivation [13]. At baseline, PDH protein was detected in all four cell lines. PDK1 and PDK2 were not detected in SK-MEL-2 cells (Figure 2B). SK-MEL-2 also showed the lowest levels of PDH phosphorylation (see also Appendix A). There was no correlation between levels of these proteins and BRAF mutational background. As expected, DCA treatment decreased PDH phosphorylation in all cell lines, while no consistent effect was seen on PDH protein levels. The degree of dephosphorylation of the three serines after DCA treatment varied, with pPDH^293^ showing the least response to DCA treatment (see Figure 2C,D and Appendix A). Since DCA is supposed to exert its effect mainly through inhibition of PDK1 and PDK2 [24], we focused on the protein levels of these two isoforms. Surprisingly, PDK2 protein levels were decreased after DCA treatment. We performed qPCR to determine whether changes in PDK2 protein levels by DCA were regulated through modulation of RNA levels (see Figure 2E,F and Appendix A). The absolute RNA levels revealed that *PDK2* in all cell lines was lower than the levels of the other *PDK* genes (Appendix A). DCA did not have a consistent effect on *PDK2* RNA levels nor *PDH* or any of the other *PDK* isoforms in the different cell lines. 

In short, DCA decreases PDH phosphorylation and protein levels of PDK2. Gene expression of *PDH* and *PDK1-4* widely varies across the different cell lines, does not correlate with protein levels and shows no clear effect following DCA treatment.

### 2.3. DCA Decreases Glycolysis in Melanoma Cells

To investigate whether the reduction in PDH phosphorylation by DCA had functional consequences for OXPHOS and glycolysis, we determined the metabolic phenotype of these cells. We first optimized DCA treatment duration, where short treatment (10 min) was sufficient to induce the same metabolic changes as 24 h treatment. DCA treatment resulted in a dose-dependent decrease in extracellular acidification rate (ECAR) and thus increased the OCR:ECAR ratio in all cell lines, albeit to varying degrees (Figure 3B,E–I and Appendix A). Oxygen consumption rate (OCR) was increased only in SK-MEL-28 cells (Appendix A). Spare respiratory capacity was decreased after DCA treatment (Figure 3C and Appendix A). DCA was previously shown to inhibit glutaminolysis [7,25]. Therefore, we performed Seahorse analysis in the presence and absence of glutamine. In both A375 and SK-MEL-28 cells, DCA treatment decreased maximum OCR and thus spare respiratory capacity in cells in the presence but not in the absence of glutamine. Glutamine availability had no effect on basal OCR, ECAR or the OCR:ECAR ratio, with or without DCA treatment (Appendix A). In summary, we demonstrate that DCA treatment decreases glycolysis (low ECAR), thereby increasing the OCR:ECAR ratio, and that glutamine presence affects the response to DCA.

### 2.4. Effect of AZD7545 Treatment on Viability, Protein Levels, RNA Levels and OCR:ECAR Ratio

In contrast to DCA, which is an allosteric inhibitor of PDK, AZD7545 binds to the lipoyl binding pocket of PDK that normally binds to the PDH complex [21]. A375 and MeWo cell lines were sensitive to AZD7545, while the SK-MEL cell lines were not. In the A375 and MeWo cell lines, the IC_50_ values were 35.0 μM and 89.3 µM, respectively. For both SK-MEL cell lines, IC_50_ of AZD7545 was not reached even with concentrations up to 100 μM (Appendix A, Table 1). AZD7545 treatment led to a decrease in PDH phosphorylation in the AZD7545 sensitive cell lines, A375 and MeWo (Appendix A), but not in AZD7545 resistant SK-MEL-28 and SK-MEL-2 cells (Appendix A). Surprisingly, AZD7545 had the opposite effect on the metabolic phenotype as was seen with DCA: both basal and maximum OCRs were dose-dependently decreased, and ECAR was increased after AZD7545 treatment, leading to a decrease in the OCR:ECAR ratio in all cell lines except SK-MEL-28 (Figure 3 vs. Appendix A). Interestingly, AZD7545 treatment resulted in an increase in PDK2 protein levels, whereas DCA decreased PDK2 levels. Since AZD7545 treatment did not lead to significant changes in RNA levels of *PDH* or *PDK1-4* (Appendix A), it is unlikely that the effect of AZD7545 on PDK2 at the protein level is transcriptionally mediated. Subsequently, we tested the effect of the proteasome inhibitor MG132 on the response to DCA and AZD7545 to determine whether differences in protein degradation were responsible for the differential effect of DCA and AZD7545 on PDK2 levels (see Appendix A). No increase in PDK2 protein levels was found after MG132 treatment. In summary, these results suggest that AZD7545-induced PDK inhibition results in various contrasting effects on metabolism, making it complicated to consider applying it as a blocker of metabolic reprogramming.

### 2.5. Effect of PDK Knockdown Is Similar to That of DCA on PDH Phosphorylation and OCR:ECAR Ratio

To investigate which isoform of PDK was mainly responsible for the effect of DCA, we generated cell lines with individual knockdowns of all four PDK isoforms. Knockdown was confirmed by PCR. Knockdown efficiency ranged from 50–80%, with PDK1 and PDK4 knockdown being the least effective and PDK3 knockdown the most effective (Figure 4A). PDK knockdown decreased PDH phosphorylation, which was most apparent for the knockdown of PDK1 and PDK3 (Figure 4B). PDK knockdown increased OCR (Figure 4C) but did not lead to a change in ECAR (Figure 4D). This resulted in an increase in the OCR:ECAR ratio for all isoforms (Figure 4E). Cell growth was slightly, but not significantly, negatively affected by PDK knockdown (Figure 4F), whereas sensitivity to DCA remained unchanged after normalization to untreated cells (Appendix A). Interestingly, knockdown of single PDKs led to an increase in OCR with little to no change in ECAR, while DCA mainly led to a decrease in ECAR and had no additional effect on the OCR in the knockdown cell lines (see Appendix A). All PDK knockdown cell lines showed an additional decrease in PDH phosphorylation after DCA treatment compared to knockdown cells without DCA, except for the PDK1 knockdown where no clear bands were visible (Figure 4B). DCA led to a further increase of the OCR:ECAR ratio in all knockdowns, showing that there is no single isoform solely responsible for the effect of DCA. These results suggest either mechanistic differences or differences in metabolic regulation between pan-PDK inhibition by DCA and individual PDK knockdown. 

### 2.6. DCA Synergizes with Other Metabolic Inhibitors

To determine whether DCA treatment leads to other metabolic co-vulnerabilities in tumor cells, we selected an array of metabolic drugs targeting different pathways in glucose metabolism and determined synergy using MTT viability assays. Differences in viability were similar when tested with a crystal violet assay. The following drugs were tested: glutaminase inhibitor CB-839 as an indirect inhibitor of TCA, monocarboxylate transporter (MCT) inhibitor diclofenac as an indirect inhibitor of glycolysis [26], metformin as a complex I inhibitor and vemurafenib as a prominent targeted treatment for BRAF-mutated melanoma, which is reported to lead to a switch to mitochondrial respiration in sensitive cells [27]. Diclofenac has been shown to inhibit MCT1 and 4 and the sensitivity of MCT4 is around ten-fold higher than that of MCT1 through binding to the transporter as well as decreasing MCT gene expression [26,28]. IC_50_ values for the metabolic drugs used here as a monotherapy are shown in Table 2. Interestingly, CB-839 was synergistic with DCA in A375, MeWo and SK-MEL-2, where around 18% growth inhibition could be attributed to synergy in MeWo and almost 32% in SK-MEL-2 (Table 3, Figure 5B and Appendix A). Synergy was not determined in SK-MEL-28 since the IC_50_ value for CB-839 was not reached with concentrations up to 1000 nM. Melanoma cells in the presence of glutamine showed reduced spare capacity after DCA treatment (Figure 3C), while no difference was seen in the absence of glutamine (Appendix A). Therefore, DCA may also influence the TCA cycle through the modulation of glutamine metabolism. Diclofenac synergized with DCA in all four cell lines with over 21% of growth inhibition attributed to synergy in MeWo and over 27% in A375 (Table 3, Figure 5 and Appendix A). Metformin showed synergy with DCA only in A375 (around 12%) and MeWo (around 23%) cells. Vemurafenib synergized with DCA in both BRAF-mutated cell lines, A375 and SK-MEL-28. 

CB-839 and diclofenac were selected for further characterization since these drugs showed high synergy with DCA in the majority of cell lines tested. Similar growth inhibitory effects of DCA, CB-839 and diclofenac alone or in combination were observed in a long-term clonogenic assay compared to the viability assay (Figure 6A). Next, we tested the combination of DCA with CB-839 or diclofenac in spheroids (Figure 6B,C). Spheroid formation was not greatly affected after treatment with monotherapy, but with both combination therapies, cells no longer formed cohesive spheroids (Figure 6B). The effect of the treatment was more pronounced when cells were treated for the full 7 days after seeding as compared to treatment of preformed spheroids only (Figure 6C). CB-839 showed a trend towards a decrease in OCR:ECAR ratio and showed a mild inhibitory effect on the OCR:ECAR ratio when combined with DCA. Diclofenac, like DCA, strongly increased the OCR:ECAR ratio, mainly through a decrease in ECAR (Figure 6D–F). The combination of DCA and diclofenac led to an even stronger increase in the OCR:ECAR ratio, showing an additive reduction in the ECAR. In summary, DCA synergized with several other metabolic drugs in reducing viability, with CB-839 and diclofenac showing the most prominent effects. Only diclofenac enhanced the effect of DCA on the OCR:ECAR ratio, corresponding to the complementary effects of these drugs on glycolysis.

## 3. Discussion

In this study, we demonstrated that DCA inhibits melanoma cell growth both in 2D and 3D cultures (Figure 1). DCA also led to a robust and dose-dependent decrease in PDH phosphorylation (Figure 2B–D), which correlated with a shift to mitochondrial respiration (Figure 3). PDK inhibition had a similar effect on PDH phosphorylation and metabolic phenotype as DCA (Figure 4). DCA synergized with several other metabolic inhibitors, showing that DCA treatment leads to targetable metabolic vulnerabilities in melanoma cells (Figure 5 and Figure 6). 

Sensitivity to DCA was independent of PDH or PDK protein levels and independent of the mutational background of the four cell lines, which represented the most common mutational backgrounds found in tumors of melanoma patients. Sensitivity to DCA was in concordance with previous studies in melanoma cell lines using DCA [14,27,29]. Since melanoma cells with acquired resistance to vemurafenib retained their sensitivity for DCA [27], these results support DCA as a potentially effective drug regardless of mutational status or sensitivity to BRAF inhibition. The inhibitory effect of DCA on glycolysis demonstrated that PDK inhibition reduces metabolic reprogramming. The decrease in PDK protein levels after DCA treatment could not be explained by differences in translation or protein stability. However, the decrease in *PDK1* and *PDK3* gene expression in MeWo cells by DCA may be caused in part through the inhibition of HIF1, as demonstrated in a recent study [30]. To gain more insight into the mechanism of action of DCA, we generated PDK1-4 knockdown cell lines. Our results demonstrated that although PDK knockdown slightly decreased cell growth, no single PDK isoform was solely responsible for the sensitivity to DCA. Although DCA is seen as a pan-PDK inhibitor, studies have shown that specificity is not equal for all PDK isoforms. The PDK1 and PDK2 isoforms are generally considered to be the most sensitive, although this remains controversial [24,31,32]. The observation that DCA can further inhibit cell growth and increase the OCR:ECAR ratio in the PDK knockdown cell lines is likely due to the presence of the other PDK isoforms. 

Surprisingly, although the exposure of cells to the PDK inhibitor AZD7545 led to decreased PDH phosphorylation in the sensitive cell lines, AZD7545 decreased the OCR:ECAR ratio in these cell lines, contrary to the effect of DCA. This is in line with previous data from a non-small cell lung cancer line [33]. These results indicate that the effect of AZD7545 on the OCR:ECAR ratio is not only dependent on the inhibition of PDH phosphorylation [21]. Our observation that PDK2 protein levels are induced by AZD7545, while a reduction in PDK2 levels was observed with DCA, suggest that changes in protein abundance of PDK isoenzymes also have functional consequences [30]. Whether the different binding sites of DCA and AZD7545 [21] play a role in PDK isoenzyme stability and abundance should be further investigated. Since we aimed to utilize PDK inhibition to decrease glycolysis, AZD7545 was not investigated further. 

As a proof-of-concept, we selected two drugs to combine with DCA, one drug that inhibits the TCA cycle by inhibiting glutaminase (CB-839) and one that reduces glycolysis by inhibiting lactate secretion via MCT1 and MCT4 (diclofenac). The Synergy of DCA with metformin or vemurafenib was previously described [27,34]. Combinations of DCA either with CB-839 or diclofenac had equal or better synergy scores as compared to DCA combined with metformin or vemurafenib. The concentrations of DCA used in the present study are higher but still in the same order of magnitude as what can be measured in the blood of patients being treated with DCA [18]. The CB-839 concentration used here was even lower than those achieved in the blood of patients treated with CB-839 [35]. Combining DCA and CB-839 might also be effective in melanoma cells resistant to BRAF inhibition, which inherently display more oxidative metabolism and glutamine dependence. BRAF inhibition-resistant cells have an increased dependence on glutamine for survival and were shown to be more sensitive to glutaminolysis inhibition by CB-839 than non-resistant cells [7]. In this case, DCA can further enhance oxidative metabolism, thereby increasing the dependency of BRAF inhibitor-resistant melanoma cells on glutamine. Therefore, CB-839 is a good candidate for combination treatment with DCA, both for BRAF inhibitor-sensitive and for BRAF inhibitor-resistant cells. The combination of DCA and CB-839 has not been tested before, but in line with our findings, the combination of DCA with another glutamine-1 inhibitor, Bis-2-(5-Phenylacetamido-1,3,4-Thia- diazol-2-yl)Ethyl Sulphide (BPTES), significantly enhanced the growth inhibitory effect of DCA in cervical and colorectal carcinoma cell lines [36]. 

To our knowledge, DCA has not been combined with diclofenac in melanoma cells previously. A potential limitation is that diclofenac was used at a concentration of around 20 times as high as those observed in the clinic [37]. Similar sensitizing effects have been observed using sulindac, another COX inhibitor, in combination with DCA in lung and squamous cell carcinoma cell lines [38]. However, changes in the metabolic phenotype after sulindac treatment were not investigated. Here, we demonstrated that the combination of DCA with diclofenac strongly reduced ECAR, suggesting inhibition of the lactate transporters MCT1 and MCT4 by diclofenac. MCT4 is upregulated during hypoxia and may therefore be of particular interest in cancer. Efforts to develop a pan-MCT or MCT4 inhibitor have been unsuccessful so far [39]; specific MCT1 inhibitors, such as AZD3965, are available but show limited efficacy [40]. Taken together, our results indicate that combination strategies with DCA can be effective. When using two metabolic inhibitors affecting the same pathway, e.g., DCA and diclofenac, we observed additive effects on the metabolic read-outs OCR and ECAR. These additive effects were not observed when targeting different metabolic pathways, e.g., glycolysis and OXPHOS. However, only acute effects of drug combinations on the metabolic read-outs were measured, and therefore, we cannot exclude that metabolic interference in time did occur. Effects on viability with both combinations were synergistic; thus, future studies will have to determine whether and how metabolic synergy underlies successful combination therapy. 

The introduction of immune checkpoint inhibitors has revolutionized the treatment of patients with metastatic melanoma, with over 50% of patients now surviving more than 5 years and large groups of patients experiencing durable responses [41]. For the group of patients not benefiting sufficiently, metabolic treatment strategies are of interest [42]. Besides direct effects on tumor cell growth, targeting cancer metabolism, especially when lactic acid secretion is decreased, as demonstrated for the combination of DCA with diclofenac, may also improve the activity of immune cells in the tumor microenvironment [43]. In murine melanoma models, CB-839 and diclofenac improved response to immunotherapy [26,44]. Therefore, the DCA combinations studied here are of interest as a potential addition to immunotherapy in melanoma. 

We focused on the protein levels of PDK1 and PDK2, which may be considered a limitation as other isoforms may also play a role in the observed effects of DCA. Moreover, we used a medium containing an abundance of glucose, amino acids and other important factors during cell culturing and experiments. This is vastly different from the conditions present in the tumor microenvironment, which is often nutrient-poor [45]. Furthermore, although the concentrations of DCA used here were comparable to the plasma levels of patients, it is unclear how these plasma levels relate to drug concentrations in tumors. Therefore, in vivo and, preferably, patient studies are needed in order to fully elucidate the relevance of these findings. 

In conclusion, PDK inhibition, through PDK knockdown and DCA treatment, leads to the reversal of metabolic reprogramming, e.g., decreased glycolysis, and leads to targetable metabolic vulnerabilities that open possibilities for DCA to be used in combination with other metabolic drugs in melanoma. Future studies should focus on determining whether DCA treatment leads to a reduction in glucose uptake and glycolysis in patients and whether it can be safely combined with other drugs. 

## 4. Materials and Methods

### 4.1. Cell Lines and Materials

Melanoma cell lines A375 (BRAF^V600E^ mutation), MeWo (BRAF/NRAS wild-type), SK-MEL-28 (BRAF^V600E^ mutation) and SK-MEL-2 (NRAS^Q61R^ mutation) were acquired from the American Type Culture Collection (ATCC, Manassas, VA, USA). Cells were cultured as monolayers according to the supplier’s instructions in RPMI-1640 medium containing 11 mM glucose (PAN Biotech #17500, Aidenbach, Germany) with 3 mM L-glutamine (ThermoFisher Scientific #25030024, Waltham, MA, USA) and 10% FBS for a maximum of 50 passages. Sodium dichloroacetate (DCA) was purchased from Sigma-Aldrich (#347795, St. Louis, MO, USA) and AZD7545 from SelleckChem (#S7517, Houston, TX, USA). DCA was dissolved in either PBS or medium (1M stock concentration) and used in concentrations between 0–50 mM. AZD7545 was dissolved in DMSO (either 10 or 100 mM stock concentration), keeping the DMSO concentration at a maximum of 0.1%, and used in concentrations between 0–100 μM. The following antibodies were used: PDH (Cell Signaling, cat.nr. 3205, Danvers, MA, USA), pPDH^Ser293^ (Calbiochem, AP1062, San Diego, CA, USA), pDPH^Ser232^ (Calbiochem, AP1063, San Diego, CA, USA), pPDH^Ser300^ (Calbiochem, AP1064, San Diego, CA, USA), PDK1 (Cell Signaling #3820, Danvers, MA, USA), PDK2 (Novus Biological, NBP2-75611, Englewood, CO, USA) and HSP90 (Santa Cruz, sc-13119, Dallas, TX, USA). 

### 4.2. Spheroid Culture

A375 cells were harvested by trypsinization and mixed with Matrigel (defrosted at 4 °C overnight; Corning #354234, Corning, NY, USA) in a 3:7 ratio of cells in medium:Matrigel, cells were resuspended in medium with a final concentration of 300 cells per 10 μL Matrigel/medium droplet. Three 10 μL droplets were placed on the bottom of a 24-well plate, after which the plate was turned upside down and placed in a 37 °C incubator. After solidifying, the medium was added either with or without DCA. Spheroids were left to grow for 7 days while replacing the culture medium with fresh (treatment) medium after 3 days. DCA was added either directly after plating (7 days before analysis), 3 days after plating (96 h before analysis) or 6 days after plating (24 h before analysis). On the day of analysis, the medium was removed and replaced by a staining solution (200 μL per well). Spheroids were stained with Hoechst (Invitrogen #H3570, Waltham, MA, USA), Calcein AM (ThermoFisher Scientific #C1430, Waltham, MA, USA) and PI (Invitrogen #P3566, Waltham, MA, USA) in the following end concentrations: 4.4 μM Hoechst, 1 μM calcein, 15 μM PI. Wells were stained for two hours at 37 °C. Subsequently, spheroids were washed twice with PBS and imaged using the ThermoFisher Scientific EVOS imaging system. In parallel, spheroids were plated in a 96-well plate with a final concentration of 900 cells in 30 μL Matrigel/medium per well. Viability was assessed via formazan formation out of 3-(4,5-dimethylthiazol-2-yl)-5-(3-carboxymethoxyphenyl)-2-(4-sulfophenyl)-2H-tetrazolium (MTS) (Promega, #G3580, Madison, WI, USA) at the end of the experiment. Subsequently, the supernatant was carefully pipetted off the Matrigel and transferred to a fresh 96-well plate prior to analysis on a plate reader at 490 nm (ThermoFisher Scientific Multiskan SkyHigh Microplate Spectrophotometer, Waltham, MA, USA).

### 4.3. Cell Viability

Cells were seeded in a density of 600–800 cells per well in a 96-well plate prior to treatment. Cells were plated, treated with the drug of interest, and left in the incubator for 96 h. The viability of the melanoma cell lines was assessed via formazan crystal formation out of 3-(4,5-dimethyl-2-thiazolyl)-2,5-diphenyl-2H-tetrazolium bromide (MTT; Sigma, St. Louis, MO, USA) after 96 h. Subsequently, the formazan crystals were dissolved in DMSO (Sigma, St. Louis, MO, USA) prior to analysis on a plate reader at 520 nm (ThermoFisher Scientific Multiskan SkyHigh Microplate Spectrophotometer, Waltham, MA, USA).

### 4.4. Immunoblotting

Cells were washed twice with PBS and lysed in Mammalian Protein Extraction Reagent (M-PER) (ThermoFisher Scientific, Waltham, MA, USA) containing protease and phosphatase inhibitors (ThermoFisher Scientific, Waltham, MA, USA). Protein content was measured by Bradford and diluted accordingly in order to load 20 µg per 20 μL sample. Samples were supplemented with Laemmli loading buffer and boiled for 10 min. ProSieve™ Color Protein Ladder (Lonza, #50550, Basel, Switzerland) was used as a reference. Samples and ladder were loaded on a 10% acrylamide gel and run on 80 to 120V for approximately 90 min. The transfer was performed using the Trans Blot Turbo system (BioRad, Hercules, CA, USA). The membrane was subsequently blocked for 60 min with 5% bovine serum albumin (BSA) or skim milk in TBST buffer. Primary antibodies were incubated overnight at 4 °C. Secondary antibodies (HRP-conjugated, DAKO, Glostrup, Denmark) were incubated for 60 min at room temperature. Lumi-Light substrate (Roche, Basel, Switzerland) was employed to visualize bands using the ChemiDoc Imaging System (BioRad, Hercules, CA, USA).

### 4.5. PCR

Cells were harvested by trypsinization, washed with PBS and RNA was extracted using the RNEasy^®^ Mini Kit (Qiagen, Hilden, Germany). RNA was then converted into cDNA using the iScript™ cDNA synthesis kit (Bio-Rad, Hercules, CA, USA). Real-time quantitative PCR was then performed using the primers as stated in Appendix A and using PowerUp SYBR Green Master Mix (ThermoFisher Scientific, Waltham, MA, USA) as a DNA dye. β-actin was used as a reference housekeeping gene.

### 4.6. Seahorse Analysis

Seahorse analysis was performed on an XFe96 Analyser from Agilent Technologies (Santa Clara, CA, USA). Cells were plated at a density of 2.0 × 10^4^ cells per well for each cell line 24 h prior to analysis. Cells were washed twice with unbuffered Seahorse medium (Agilent #102353-100 DMEM, Santa Clara, CA, USA, with or without 3 mM L-glutamine and with 10 mM glucose) prior to adding Seahorse medium containing drugs. Cells were treated with the drugs for 10 min prior to starting the assay, during which the drugs remained present. Oxygen consumption rate (OCR) and extracellular acidification rate (ECAR) were measured during subsequent injections of 2.5 µM oligomycin (ATP synthase inhibitor, Sigma #O4876, St. Louis, MO, USA), 50 µM 2,4-dinitrophenol (DNP; uncoupler, Sigma #D19850, St. Louis, MO, USA), 2 µM rotenone with 4 µM Antimycin A (complex I and III inhibitors, respectively, Sigma #R8875 and #A8674, St. Louis, MO, USA) and lastly, 100 mM 2-deoxyglucose (2-DG; glucose analog and hexokinase II inhibitor, Sigma #D8375, St. Louis, MO, USA); these are final concentrations in the wells. The OCR:ECAR ratio was calculated from basal OCR and ECAR corrected for background levels of OCR and ECAR after 2-DG injection. Spare capacity was calculated by subtracting the basal OCR from the maximal OCR.

### 4.7. Generation of PDK Knockdown Cell Lines

The lentiviral transduction of short hairpin RNA (shRNA) was used to inhibit the expression of the target genes. Sequences for each shRNA construct can be found in Appendix A. Lentiviruses containing a pRRL.SFFV.EGFP backbone containing shRNA constructs against either PDK1-4 or luciferase were used, with the shRNA targeting luciferase as a control. Cells were sorted based on GFP expression after transduction on a MoFlow XDP cell sorter (Beckman Coulter, Brea, CA, USA) to ensure a ~100% transduced cell population.

### 4.8. Combination Treatments

The following drugs were used in combination with DCA: CB-839 (SelleckChem, #S7655, Houston, TX, USA, 0–200 nM), diclofenac (ICN Biomedicals, #15307-79-6, Costa Mesa, CA, USA, 0–200 µM), vemurafenib (Axon Medchem, #1624, Reston, VA, USA, 0–240 nM) and metformin (Sigma-Aldrich, #D150959, St. Louis, MO, USA, 0–10 mM). Glycolysis was blocked indirectly through diclofenac (cyclooxygenase-2 (COX-2) and MCT inhibitor) by blocking lactate efflux. The TCA cycle was blocked indirectly with metformin (complex I inhibitor) by blocking respiration and by CB-839 (glutaminase inhibitor) by blocking the utilization of glutamine for the TCA cycle. In addition, we used the BRAF-inhibitor vemurafenib, which is used clinically. Single drug IC_50_ values were determined prior to testing the drugs in combination with DCA. Cells were treated with the combination of two drugs for 96 h prior to viability analysis using formazan crystal formation as a readout (see Section 4.3). Synergy scores were calculated using the SynergyFinder online tool developed by Ianevski et al. [46]. Synergy is assumed with values above 10, where the value represents the average excess response of two drugs, i.e., with a synergy score of 20, 20% of the response can be attributed to drug interactions and not addition alone.

### 4.9. Statistical Analysis

Data are expressed as mean ± SD (standard deviation). At least three independent experiments were performed, each consisting of at least three technical replicates, as indicated in the figure legends. Normal distribution was tested using the Shapiro–Wilk test. Data were considered to be statistically significant when *p* < 0.05. Different experimental conditions were tested using Student’s *t*-test or one-way ANOVA followed by Dunnett’s multiple comparison test, as indicated in the figure legends. All data were analyzed with the GraphPad Prism 8.0 software.

## Figures and Tables

**Figure 1 ijms-23-03745-f001:**
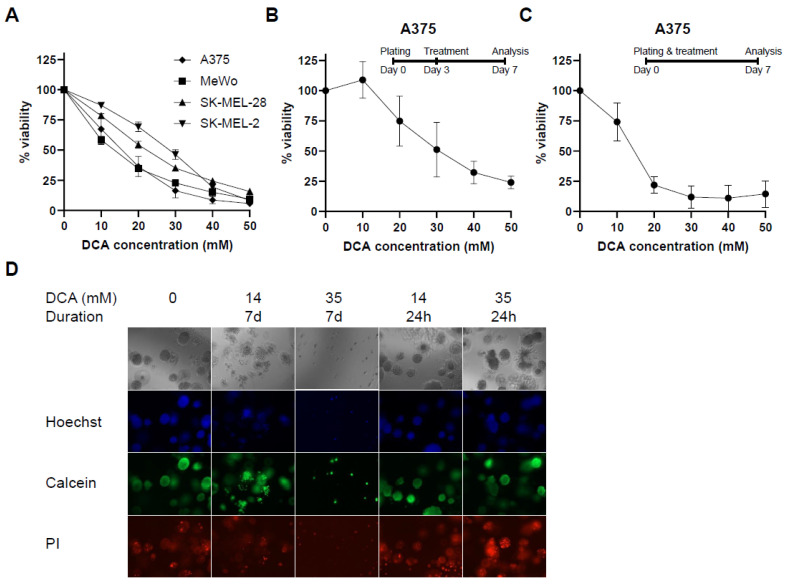
Effect of PDK inhibition on melanoma cell proliferation. (**A**) Viability curves after 96 h of treatment with DCA in all cell lines. Viability curve of A375 cells grown in Matrigel treated with DCA for either (**B**) 96 h after 3 days of initial spheroid development or (**C**) treated continuously for 7 days. (**D**) Representative image of Hoechst/Calcein/PI staining of A375 spheroids treated with 14 mM DCA either continuously for 7 days or 24 h after 6 days of initial spheroid formation. 10× magnification. (**A**–**C**) Data represent mean ± SD of 3 independent experiments, each performed in triplicate. PI: propidium iodide.

**Figure 2 ijms-23-03745-f002:**
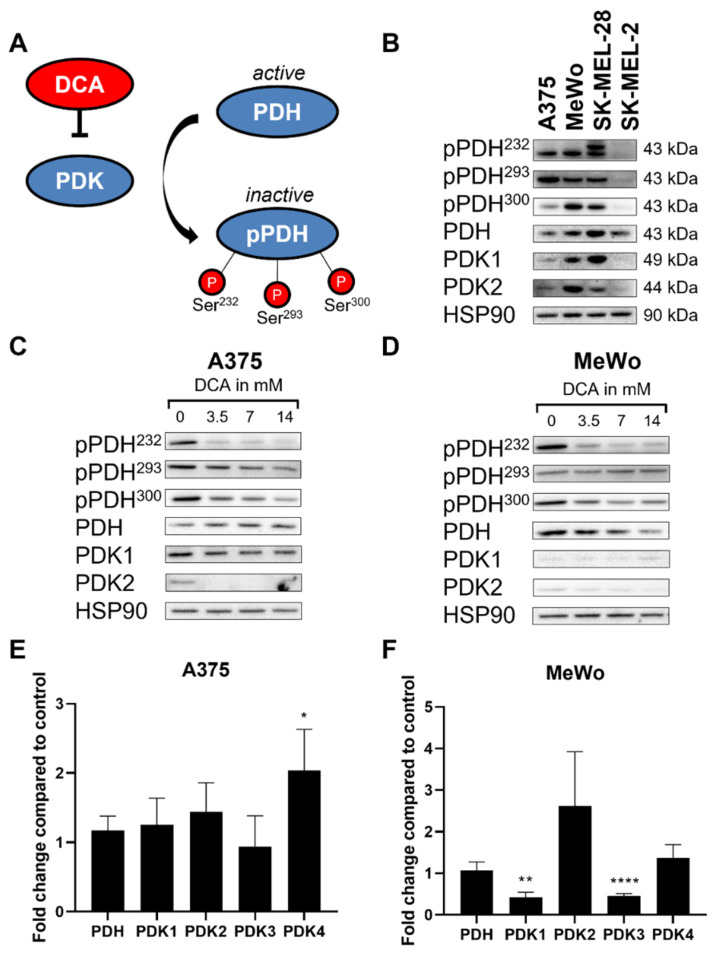
Effect of PDK inhibition on PDH and PDK levels in melanoma cells. (**A**) Schematic view of PDH/PDK axis. (**B**) Representative image of protein levels of metabolic enzymes at baseline with no treatment in all four cell lines. Representative image of protein levels after 24 h treatment with DCA in (**C**) A375 and (**D**) MeWo. Fold change in RNA levels of *PDH* and *PDK1-4* after 24 h treatment with DCA compared to control in (**E**) A375 and (**F**) MeWo. (**E**,**F**) Data represent mean ± SD of 3 independent experiments, each performed in quadruplicate. Student’s *t*-test, * *p* < 0.05, ** *p* < 0.01, **** *p* < 0.0001, DCA-treated vs. control.

**Figure 3 ijms-23-03745-f003:**
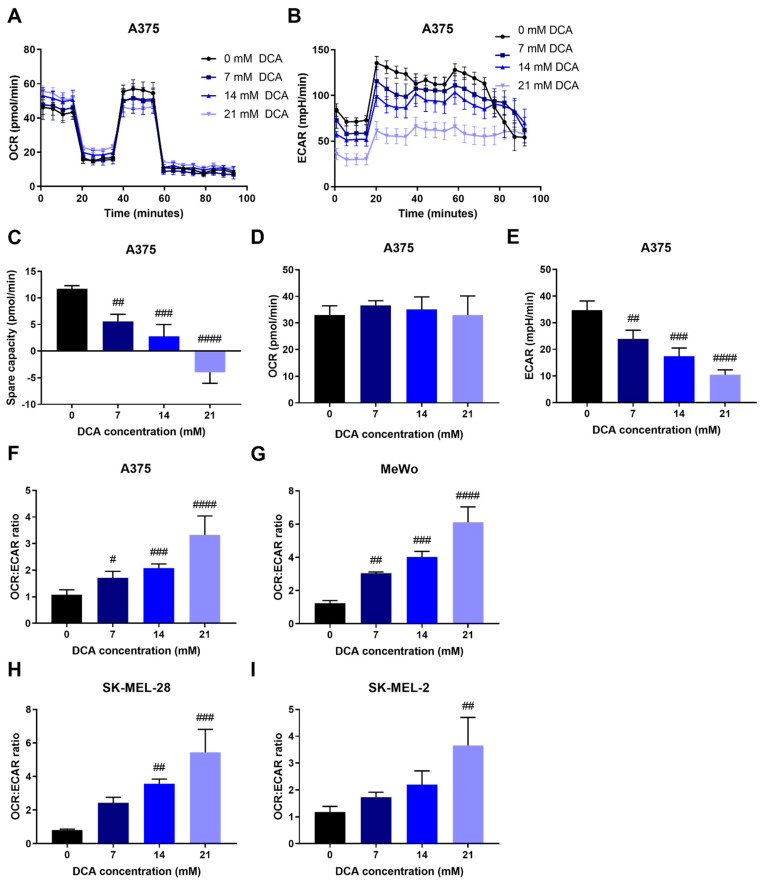
Metabolic adaptation upon PDK inhibition. (**A**) Representative image of OCR in A375 cells measured by Seahorse, mean ± SD of 6–8 technical replicates are shown. (**B**) Representative image of ECAR in A375 cells measured by Seahorse, mean ± SD of 6–8 technical replicates are shown. (**C**) Spare capacity, e.g., maximal OCR minus basal OCR, is decreased by DCA treatment in A375 cells. DCA treatment does not affect (**D**) basal OCR, whereas it dose-dependently decreases (**E**) ECAR in A375 cells. DCA treatment increases the OCR:ECAR ratio in (**F**) A375, (**G**) MeWo, (**H**) SK-MEL-28 and (**I**) SK-MEL-2 cells. (**C**–**I**) Data represent mean ± SD of 3–6 independent experiments, each consisting of 6–8 technical replicates. One-way ANOVA followed by Dunnett’s multiple comparison test, # *p* < 0.05, ## *p* < 0.01, ### *p* < 0.001, #### *p* < 0.0001, DCA-treated vs. control.

**Figure 4 ijms-23-03745-f004:**
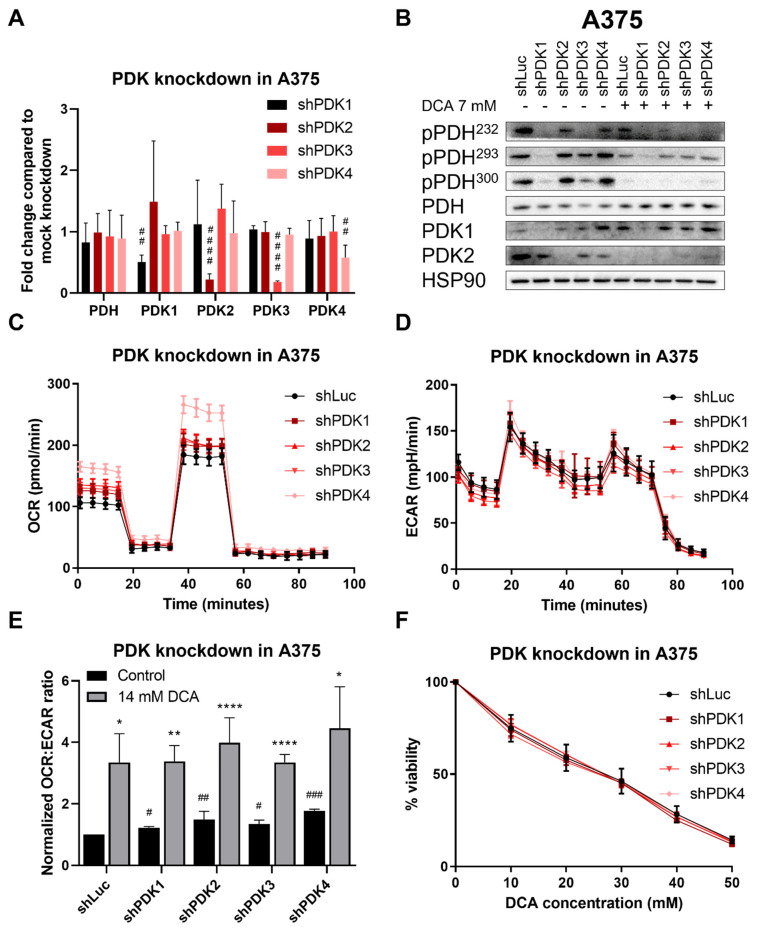
Effect of PDK knockdown on RNA levels, protein levels and viability under DCA treatment. (**A**) Fold change in RNA levels of *PDH* and *PDK1-4* after PDK knockdown compared to the mock knockdown. Data represent mean ± SD of 3 independent experiments, each performed in quadruplicate. One-way ANOVA followed by Dunnett’s multiple comparison test, ## *p* < 0.01, #### *p* < 0.0001, shPDK1-4 vs. shLuc. (**B**) Representative image of protein levels after PDK knockdown and/or DCA treatment. (**C**) Representative image of OCR in PDK knockdown cell lines measured by Seahorse, mean ± SD of 6–8 technical replicates are shown. (**D**) Representative image of ECAR in PDK knockdown cell lines measured by Seahorse, mean ± SD of 6–8 technical replicates are shown. (**E**) PDK knockdown increases the OCR:ECAR ratio, whereas OCR:ECAR ratio is even further increased after DCA treatment in all knockdowns. Data represent mean ± SD of 3 independent experiments, each consisting of 6–8 replicates. One-way ANOVA followed by Dunnett’s multiple comparison test, # *p* < 0.05, ## *p* < 0.01, ### *p* < 0.001, shPDK1-4 vs. shLuc (black bars). Student’s *t*-test * *p* < 0.05, ** *p* < 0.01, **** *p* < 0.0001, DCA-treated vs. control of the same shRNA (grey bars). (**F**) The sensitivity of A375 cells to DCA does not change after PDK knockdown. Data are mean ± SD of 3 independent experiments, each performed in triplicate.

**Figure 5 ijms-23-03745-f005:**
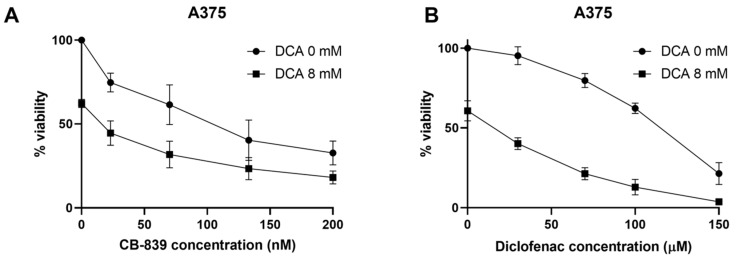
DCA shows synergy with other metabolic inhibitors. (**A**) Viability curve of A375 cells after 96 h of treatment with CB-839 with and without DCA. (**B**) Viability curve of A375 cells after 96 h of treatment with diclofenac with and without DCA. Data represent mean ± SD of 3 independent experiments, each performed in triplicate.

**Figure 6 ijms-23-03745-f006:**
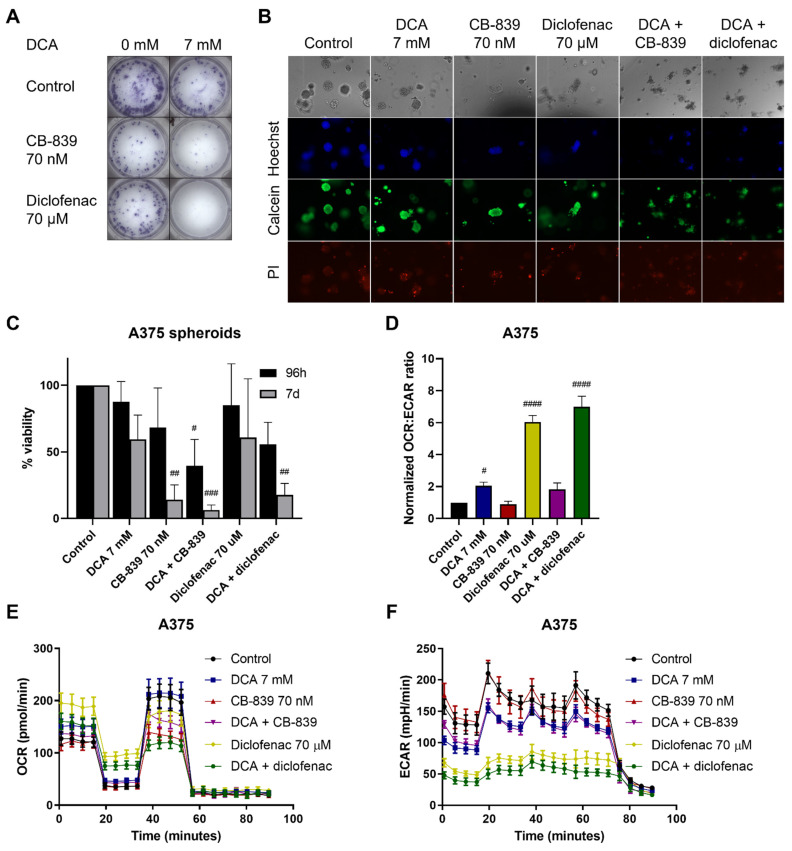
Clonogenicity and metabolic phenotype of A375 cells under DCA treatment combined with CB-839 and diclofenac. (**A**) Representative image of clonogenic assay of A375 after 9 days of treatment with DCA and CB-839 or diclofenac. (**B**) Representative image of Hoechst/Calcein/PI staining of A375 spheroids treated with DCA and CB-839 for 96 h. 10× magnification. (**C**) Viability of A375 cells grown in Matrigel treated with DCA and CB-839 or diclofenac for either 96 h after initial spheroid development or treated continuously for 7 days. (**D**) OCR:ECAR ratio (here, normalized to untreated cells) after DCA, CB-839, diclofenac and combination treatments. (**E**) Representative image of OCR in A375 cells measured by Seahorse, mean ± SD of 6–8 technical replicates are shown. (**F**) Representative image of ECAR in A375 cells measured by Seahorse, mean ± SD of 6–8 technical replicates are shown. (**C**,**D**) Data represent mean ± SD of 3 independent experiments, each performed in quadruplicate. One-way ANOVA followed by Dunnett’s multiple comparison test, # *p* < 0.05, ## *p* < 0.01, ### *p* < 0.001, #### *p* < 0.0001, drug-treated vs. control.

**Table 1 ijms-23-03745-t001:** IC_50_ values for DCA and AZD7545 *.

Cell Line	DCA IC_50_ (mM)	AZD7545 IC_50_ (μM)
A375	14.9 ± 1.0	35.0 ± 4.1
MeWo	13.3 ± 0.6	89.3 ± 5.0
SK-MEL-28	20.0 ± 1.4	>100
SK-MEL-2	27.3 ± 1.7	>100

* Cells were treated for 96 h with either DCA or AZD7545, after which IC_50_ values, e.g., the concentration at which cells had 50% viability compared to the untreated cells, were calculated from the resulting viability curve. Data represent mean ± SD of 3–6 independent experiments, each performed in triplicate.

**Table 2 ijms-23-03745-t002:** IC_50_ values for monotherapy with CB-839, diclofenac, metformin and vemurafenib *.

Cell Line	CB-839 IC_50_ (nM)	Diclofenac IC_50_ (μM)	Metformin IC_50_ (mM)	Vemurafenib IC_50_ (nM)
A375	99.4 ± 37.7	112.7 ± 4.1	3.0 ± 0.8	70.7 ± 22.7
MeWo	7.9 ± 0.9	146.0 ± 27.0	7.5 ± 2.6	N/A
SK-MEL-28	N/A	134.0 ± 4.6	N/A	82.0 ± 25.4
SK-MEL-2	139.2 ± 34.8	133.8 ± 13.6	N/A	N/A

* Cells were treated for 96 h with the drug in question, after which IC_50_ values, e.g., the concentration at which cells had 50% viability compared to the untreated cells, were calculated from the resulting viability curve. Data represent mean ± SD of 3–6 independent experiments, each performed in triplicate. N/A, not applicable.

**Table 3 ijms-23-03745-t003:** Synergy scores of the tested combinations in all cell lines *.

Drug	A375	MeWo	SK-MEL-28	SK-MEL-2
CB-839	20.3	17.7	N/A	31.7
Diclofenac	27.4	21.5	25.1	27.0
Metformin	12.4	23.1	N/A	N/A
Vemurafenib	26.0	N/A	23.8	N/A

* Synergy is assumed with values above 10, where the value represents the average excess response of two drugs, i.e., with a synergy score of 20, 20% of the response can be attributed to drug interactions and not addition alone. Tabs with “N/A” indicate that cell lines were insufficiently sensitive to the drug in question to determine synergy. Each synergy score is based on data from 3 independent experiments, each performed in triplicate.

## Data Availability

Not applicable.

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
