# Peer review of "Pyruvate Dehydrogenase Kinase Inhibition by Dichloroacetate in Melanoma Cells Unveils Metabolic Vulnerabilities"

_ijms, 2022, doi:10.3390/ijms23073745_

Round 1

Reviewer 1 Report

The authors study pyruvate dehydrogenase kinase inhibition by dichloroacetate in melanoma cells. Identification of metabolic modulators reducing the viability of melanoma cells is a very current topic. Some metabolic modulators, including the pyruvate dehydrogenase kinase inhibitor dichloroacetate (DCA) and the biguanide metformin, sensitize melanoma cells to BRAF inhibitors.

The authors should indicate from the beginning in the abstract that it is a cellular model (only talking about melanoma cells misleads the reader, although from the context we know that they are).

In the statistical analysis, the authors indicate that they have used the mean as the mean of central tendency and the standard deviation as a measure of dispersion and the Student's t-test to compare the means. However, they do not indicate whether the distribution of the variables is similar to normal, and therefore allows the use of parametric tests. The value of p does not appear in any of the graphs either.

Limitations of the study are not stated.

Author Response

Response to Reviewer 1 Comments

We thank Reviewer 1 for the comments and appreciate that Reviewer 1 highlights the relevance of our study in the current field of melanoma treatment.

  1. The authors should indicate from the beginning in the abstract that it is a cellular model (only talking about melanoma cells misleads the reader, although from the context we know that they are).

The reviewer points out that this may be a point of confusion. Accordingly, we changed our wording in the abstract and introduction (page 1, line 14 and 23 and page 2, line 78).

  1. In the statistical analysis, the authors indicate that they have used the mean as the mean of central tendency and the standard deviation as a measure of dispersion and the Student's t-test to compare the means. However, they do not indicate whether the distribution of the variables is similar to normal, and therefore allows the use of parametric tests. The value of p does not appear in any of the graphs either.

The reviewer makes a valid point. We have now tested for normal distribution using the Shapiro-Wilk test and all our data was normally distributed, which allowed us to utilize parametric tests. We have now included this in our methods section (page 16, line 534). We also changed the test method from the Student’s t-test to a one-way ANOVA followed by Dunnett’s multiple comparison test in the case of comparison of multiple groups (page 16, lines 537-539). We changed all figure legends and supplementary figure legends accordingly. Our data analysis and interpretation changed slightly due to some small differences in p values (page 4, line 144; page 6, line 190; page 6, lines 201-202; page 10, lines 288-290). The omission of the value of p was an oversight on our part and we thank the reviewer for pointing this out to us. We have included the value of p in the figure legends.

  1. Limitations of the study are not stated.

We agree with the reviewer that the limitations of our study were not adequately described and we have therefore included a paragraph describing the limitations at the end of the discussion (page 13-14, lines 416-427).

Reviewer 2 Report

Manuscript can be accepted as it is.

Author Response

Response to Reviewer 2 Comments

We thank Reviewer 2 for their positive response to our manuscript.

Round 2

Reviewer 1 Report

The authors have adequately responded to all suggestions from this reviewer.

Author Response

We thank Reviewer 1 for their positive response to our manuscript.